# A penalized variable selection ensemble algorithm for high-dimensional group-structured data

**Dongsheng Li**[1,2]*, **Chunyan Pan**[1], **Jing Zhao**[1], **Anfei Luo**[3]*

**1** School of Mathematics and Statistics, Qiannan Normal University for Nationalities, Duyun, Guizhou, China, **2** Key Laboratory of Complex Systems and Intelligent Optimization of Guizhou Province, Duyun, China, **3** Department of Computer Science, Guizhou Police College, Guiyang, Guizhou, China

\* 465164730@qq.com (DL); laf911205@163.com (AL)

**Data Availability Statement:** All relevant data are within the manuscript and its Supporting Information files.

**Funding:** This work was supported by The Guizhou Provincial Department of Education's Youth Growth Project Fund [No. Qian Jiaoji [2022] 380,

## Abstract

This paper presents a multi-algorithm fusion model (StackingGroup) based on the Stacking ensemble learning framework to address the variable selection problem in high-dimensional group structure data. The proposed algorithm takes into account the differences in data observation and training principles of different algorithms. It leverages the strengths of each model and incorporates Stacking ensemble learning with multiple group structure regularization methods. The main approach involves dividing the data set into $K$ parts on average, using more than 10 algorithms as basic learning models, and selecting the base learner based on low correlation, strong prediction ability, and small model error. Finally, we selected the grSubset + grLasso, grLasso, and grSCAD algorithms as the base learners for the Stacking algorithm. The Lasso algorithm was used as the meta-learner to create a comprehensive algorithm called StackingGroup. This algorithm is designed to handle high-dimensional group structure data. Simulation experiments showed that the proposed method outperformed other $R^2$, $RMSE$, and $MAE$ prediction methods. Lastly, we applied the proposed algorithm to investigate the risk factors of low birth weight in infants and young children. The final results demonstrate that the proposed method achieves a mean absolute error ($MAE$) of 0.508 and a root mean square error ($RMSE$) of 0.668. The obtained values are smaller compared to those obtained from a single model, indicating that the proposed method surpasses other algorithms in terms of prediction accuracy.

## 1 Introduction

Data has recently evolved into a brand-new category of crucial manufacturing components. The study of image processing, text mining, and genetic informatics has progressively advanced with the rapid growth of artificial intelligence. Large-scale database mining and analysis are starting to gain more and more attention. The complexity of the objects being processed increases along with the complexity of the data dimensions. The number of samples that can be utilized for analysis is too few when compared to ultra-high-dimensional data, resulting in high-dimensional small sample data. Data having a particularly high data

No. Qian Jiaoji [2022] 377, No. Qian Jiaoji [2022] 378, and No. Qian Jiaohe KY [2021] 282]; and in part by the Educational Department of Guizhou under Grant [No.KY[2019]067].

**Competing interests:** The authors have declared that no competing interests exist.

dimension, a tiny absolute number of samples, or a significantly lower number of samples than the data dimension are referred to as "high-dimensional small sample data." High-dimensional small sample data will seriously devastate the data analysis process in terms of dimension. A small sample size makes it simple to overfit the model, which makes it tough to extrapolate and predict and also makes choosing the right variables challenging. Variable selection, which has long been a crucial component of statistical modeling, is a useful approach to resolving this issue. Variable selection is a technique used to improve statistical models by deleting duplicate or pointless variables. Since it was first introduced in the 1960s, variable selection theory and methodology have been a significant topic in statistical study. Initially, the sample size is adequate since the sample dimension of the data set is frequently less than 40. Thus, the subset technique and coefficient compression method are the most often utilized methods. The area of statistics makes extensive use of these methods. The flaws in these algorithms are starting to become apparent as the big data age begins. Tibshirani (1996) [1] introduced Lasso (Least Absolute Shrinkage and Selection Operator), which was modeled after the NG (Nonnegative Garrote) approach, to address the drawbacks of conventional statistical techniques. In order to actualize the sparsity of the model and variable selection, the model solves the gradient at the zero point using the $L_1$ norm penalty to get a sparse solution in which many of the components of the model vector are zero. Stepwise regression, the optimum subset, and model selection are examples of procedures that it subverts that are greedy. A compressed angle is used to accomplish automatic recognition. Since then, penalty-based variable selection methods have emerged one after another (Breiman [2], 1996; Tibshirani [3], 2011), aiming at the shortcoming of Lasso's biased estimation, a series of approximate unbiased sparse models have been proposed one after another. The so-called approximate unbiased sparse model refers to the sparse model that weakens or even eliminates the compression of the regression coefficient of the target variable compared with Lasso. These approximate unbiased sparse models are Bridge (Frank and Friedman, 1993), SCAD (Fan and Li [4], 2001; smoothly Clipped Absolute Deviation), adaptive lasso (Zou [5], 2006; alasso), MCP (Zhang [6], 2007), $L_2$ SCAD (Jun [7], 2014), etc. The above methods are all methods for selecting a single variable; however, it is common for explanatory variables to possess a group structure. For instance, when analyzing gene expression, genes that belong to the same biological pathway or share similar biological functions can be regarded as a group. Neglecting this grouping information during data analysis can significantly diminish the impact of variable selection, decreasing the model's explanatory power. As a result, researchers have begun to explore methods of variable selection. Hui (2005) [8] conducted a study on this topic, while Yuan and Lin (2006) [9] proposed the Group Lasso model, which utilizes the group structure between variables as prior information. One advantage of this model is that its objective function is a convex function of unknown parameters, ensuring the existence of a unique global minimum. Numerous scholars have researched the properties of this model. Group Lasso, being based on the $L_1$ norm penalty, shares similar disadvantages with Lasso. Specifically, it tends to overly compress groups with large coefficients, leading to significant parameter estimation deviation and an excessive selection of groups. To address these issues, the Adaptive Group Lasso was proposed by Wang (2007) et al. [10]. Additionally, Group Lasso's estimation is biased, prompting the development of alternative models. The Group SCAD model utilizes the SCAD penalty function (Wang, 2007 [11]; Hai, 2015 [12]), while the Group MCP model employs the MCP penalty function (Huang, 2012 [13]). Moreover, the Sparse Group Lasso (Simon, 2013) [14] is a combination of Lasso and Group Lasso for a two-layer variable selection. Its optimal solution is determined by its first-order derivative function, which guarantees a global optimal solution without the presence of local optimal solution problems common in optimization. Later on, alternative approaches to the Group Lasso algorithm were introduced in the literature. Fang

[15] (2015) devised the adaptive sparse group Lasso method, while Jung [16] (2015) proposed the Bayesian Sparse Group Selection technique. To address the limitation of Group Lasso in selecting variables within a group, Jian [17] (2009) proposed the $L_1$ Group Bridge method. Additionally, Breheny and Huang [18] (2009) suggested the $L_1$ Group MCP approach to tackle the non-differentiability issue of the $L_1$ Group Bridge penalty function. Furthermore, Breheny [19] (2015) explored the Group Lasso, Group MCP, and Group SCAD algorithms, and extended these algorithms to logistic regression.

In various research and application domains, the increasing prevalence of vast high-dimensional data underscores the significance of leveraging the sparse characteristics of the data to extract valuable insights. Trevor (2020) [20] enhanced and expanded the simulation study on optimal subset selection, while also conducting comparisons with forward stepwise selection, Lasso, and relaxed Lasso. Moreover, there have been advancements in the Best subset selection algorithm by Guo (2020) [21], as well as extensions to group variable selection by Guo [22] (2014) and Zhang [23] (2020). In recent years, ensemble learning has garnered significant attention in the field of machine learning due to its ability to efficiently solve practical application problems. Ensemble learning methods involve training multiple learners and combining their outputs to address a problem. Some of the most well-known ensemble learning methods include Bagging, Boosting, and Stacking. These methods have demonstrated superior accuracy compared to single learners, leading to great success in various practical tasks. In recent years, due to ensemble learning (Dasarathy and Sheela [24], 1979; Jacobs [25], 1991) can efficiently solve practical application problems, so it has attracted much attention in the field of machine learning. These methods train multiple learners and combine them to solve a problem. The most classical ensemble learning methods are Bagging (Breiman [26], 1996), Boosting (Schapire [27], 1998; Freund and Schapire [28], 1997) and Stacking (Wolpert [29], 1992). In general, an ensemble algorithm combining multiple learners is more accurate than a single learner, and the ensemble learning method has achieved great success in many practical tasks (Anders [30], 1997; Hansen [31], 2002; Xin [32], 2010; Gras [33], 2017). In the realm of statistics, the focus of research has always been on the method that serves as the basis for selecting variables, as each method yields different selections. To arrive at a more effective solution strategy, it is advisable to employ multiple commonly used variable selection methods separately and integrate the common factors among these methods. Notably, some scholars have expanded the concept of ensemble learning to include group variable selection algorithms (Wang [34], 2022; Wan and Tanioka [35], 2023; Hussein and Rahul [36], 2023). To accommodate more complex group structures, Thompson (2021) [37] proposed a sparse estimator of group structure by combining group subset selection and shrinkage. As a result, algorithms such as grSubset + grLasso and grSubset + Ridge were developed to enable the dual-layer selection of group variables.

In view of the advantages of regularized sparse model and ensemble learning, this paper proposes a multi-algorithm fusion model based on the Stacking ensemble learning framework, which is named StackingGroup. The algorithm idea is as follows: first, the algorithm that can select a single variable and a double-layer selection variable is used as the base learner; then, through simulation experiments, the base learner with good adaptability to high-dimensional small sample group structure data and high model fitting degree is selected as the optimal base learner. Finally, the combination strategy of Stacking ensemble learning is adopted, and a layer of the Lasso model is used as a meta-learner to train the data and output the prediction results. Through simulation experiments and real data verification, the results show that compared with a single model, the method proposed in this paper has certain generalization and effectiveness for high-dimensional small sample and large sample group structure data, and provides a feasible research framework for processing high-dimensional group structure data.

The subsequent sections of this article are organized as follows: The second section provides an overview of the fundamental components of the model. In the third section, a series of simulation experiments are conducted, employing various methodologies for comparative analysis and selecting the base learner. The fourth section applies the proposed method to real-world problems characterized by high dimensionality and limited sample sizes, validating the effectiveness of the model in terms of prediction. Finally, the fifth section offers a comprehensive summary of the entire text.

## 2 Algorithm principle

### 2.1 Model formulation

Consider the following general linear model with a group structure

$$Y_{n \times 1} = \sum_{g=1}^{G} X_g \boldsymbol{\beta}_g + \boldsymbol{\varepsilon} \tag{1}$$

Where $X_g$ is $n \times m_g$ the dimension of the design matrix, $\beta_g$ is a regression coefficient vector of length $m_g$, and $\varepsilon_{n \times 1} \sim N_n(0, \sigma^2 I_n)$, Number of non-overlapping groups $g = 1, 2, \cdots, G$, and the dimension is $p = \sum_{g=1}^{G} m_g$.

### 2.2 Group lasso

To solve the variable selection of high-dimensional variables, the Group Lasso (grLasso) method has been proposed and widely used by many scholars (Yuan and Lin [9], 2006; Geer [39], 2008), the specific model expression is as follows

$$\min_{\beta} \left\| Y - \sum_{g=1}^{G} X_g \boldsymbol{\beta}_g \right\|_2^2 + \lambda \sum_{g=1}^{G} \|\boldsymbol{\beta}_g\|_2 \tag{2}$$

### 2.3 Group SCAD

Consider the linear model (See Eq 1), assuming that $m_g$ variables in the model are divided into g non-overlapping groups, Group SCAD (Wang, 2007 [11]; Hai [12], 2015; the solution of the objective function of the grSCAD model is

$$\min_{\beta} \left\| Y - \sum_{g=1}^{G} X_g \boldsymbol{\beta}_g \right\|_2^2 + \sum_{g=1}^{G} P_{\lambda_n}(\|\boldsymbol{\beta}_g\|_2) \tag{3}$$

where the penalty function $P_\lambda(t)$ is defined as

$$P_\lambda(t) = \lambda \left\{ I(t \leq \lambda) + \frac{(a\lambda - t)}{(a-1)\lambda} I(t > \lambda) \right\}$$

where $t > 0$ in the penalty function, $\lambda > 0$ is a tuning parameter, $a > 2$ is a parameter, and the subscript $n$ is used to indicate that $\lambda$ is a function of sample size.

### 2.4 Group subset

The general Lasso and Group Lasso algorithms are not suitable for overlapping groups; if two groups are overlapping, the algorithm cannot select a variable independently of the other group; To solve this problem, a group-specific vector needs to be introduced $\overline{\boldsymbol{v}}_k \in R^p (k = 1, 2, \cdots, g)$, Except for the position indexed by $G_k$, the rest places are zero. Let $V$

be the set $\overline{\boldsymbol{v}} := (\overline{\boldsymbol{v}}_1, \cdots, \overline{\boldsymbol{v}}_g)$ of all tuples, then (Thompson [37], 2021) the objective function with Group subset is

$$\min_{\beta \in R^p, \overline{\boldsymbol{v}} \in V} \left\| Y - \sum_{g=1}^{G} X_g \boldsymbol{\beta}_g \right\|_2^2 + \sum_k \lambda_0 1(\|\overline{\boldsymbol{v}}_k\| \neq 0) + \sum_k \lambda_{1k} \|\overline{\boldsymbol{v}}_k\| \qquad (4)$$
$$\beta = \sum_k \overline{\boldsymbol{v}}_k$$

where the group-specific vector $\overline{\boldsymbol{v}}_k$ is a decomposition of the regression coefficients $\boldsymbol{\beta}$ into a sum of potential coefficients, facilitating the selection of overlapping groups, and $\lambda$ is a tuning parameter.

## 2.5 Coordinate descent algorithms

Coordinate Descent Algorithms (CDA) belong to a category of optimization algorithms used to address the task of locating the minimum or maximum value within a function space. The fundamental concept behind CDA involves updating a single variable (coordinate) during each iteration while keeping the remaining variables unchanged. More precisely, the coordinate descent algorithm follows the subsequent steps in its iterative process:

- Initialization variable: given the initial solution vector;

- Select Variables to Update: Select a variable to be updated from all variables;

- Updating selected variables: Updating selected variables to new values that reduce the objective function while leaving other variables unchanged;

- Check the convergence condition: determine whether the stopping criterion is satisfied, and if so terminate the algorithm; otherwise, return to step 2 for the next iteration;

- Output optimal solution: When the stopping criterion is reached, the optimal solution is output.

The coordinate descent method only takes into account the updating of one variable every iteration, choosing the variable to be updated in accordance with a predetermined rule. This rule can be chosen heuristically, sequentially, or at random.

When the objective function can be written as the sum of subfunctions in each dimension, a property known as coordinate separation, the coordinate descent technique can be used to solve the issue. By independently improving each subfunction, it is possible to slowly approach the global optimal solution in this situation.

Algorithm 1 below lists the main steps of the Coordinate descent algorithm for the $\boldsymbol{v}^{(m)}$ vector update iteration

---

**Algorithm 1:Coordinate descent**

**Input:** $\boldsymbol{v}^{(0)} \in R^{\sum_k p_k}$
**for** $m = 1, 2, \cdots do$
 $\boldsymbol{v}^{(m)} \leftarrow \boldsymbol{v}^{(m-1)}$
**for** $k = 1, \cdots, g\ do$
$\boldsymbol{v}_k^{(m)} \leftarrow argmin_{\xi \in R^{p_k}} \overline{F}_{ck}(\boldsymbol{v}_1^{(m)}, \boldsymbol{v}_2^{(m)}, \cdots, \boldsymbol{v}_{k-1}^{(m)}, \xi, \boldsymbol{v}_{k+1}^{(m)}, \cdots, \boldsymbol{v}_g^{(m)}; \boldsymbol{v}^{(m)})$
end
if converged **then break**
end
return $\boldsymbol{v}^{(m)}$

---

**Proposition 1** *Suppose that $\hat{v}^{(t)}$ is the result of running Algorithm 1 with $\lambda_0 = \lambda_0^{(t)}$. Let $A^{(t)}$ be the active set of groups. Then running Algorithm 1 initialized to $\hat{v}^{(t)}$ and using $\lambda_0 = \lambda_0^{(t+1)}$ where*

$$\lambda_0^{(t+1)} = \alpha \cdot \max_{k \notin A^{(t)}} \left( \frac{\left( \|\nabla L(\hat{v}^{(t)})\| - \lambda_{1k} \right)_+^2}{2 p_k \bar{c}_k} \right)$$

*produces a solution $\hat{v}^{(t+1)}$ such that $\hat{v}^{(t+1)} \neq \hat{v}^{(t)}$ for any $\alpha \in [0,1]$.*

Table 1 summarizes the penalty functions and estimates for the other methods used in this paper.

## 2.6 Stacking

In order to combine the prediction outcomes of many base models for improved overall performance, Wolpert [29] created the Ensemble Learning (EL) technique of stacking in 1992. It is a meta model that takes inputs from other base models' predictions and uses them to produce final predictions.

In Stacking, there are two main components: The Base Models and the Meta Model.

- Base Models: Base models are multiple separate predictive models that are trained independently on the training data. These models can be different types of models or models of the same type but trained with different features, parameters, or random initializations. Each base model makes predictions on the training data and generates an independent prediction.

- Meta model: The meta model is a model used to combine the prediction results of the basic model to generate the final prediction results. It accepts the prediction results of the basic model as input and generates the final prediction based on their weights or probabilities.

The basic process of Stacking is as follows:

- Training the base model: The individual base models are trained using the training data.

- Generate predictions for base models: For training data, each base model generates a separate prediction.

**Table 1. Penalty functions and estimators for some of the regularized regression methods used in this paper.**

| Method | Penalty | Estimator |
|---|---|---|
| grMCP [13] | $P_\lambda(\beta) = \sum_{g=1}^{G} f_{\lambda,b}^{MCP}\left( \sum_{k=1}^{p_j} f_{\lambda,b}^{MCP}(|\beta_k^{(j)}|) \right)$ | $\hat{\beta}_{grMCP} = argmin\{\|\boldsymbol{y} - \boldsymbol{X\beta}\|_2^2 + P_\lambda(\beta)\}$ |
| lasso [1] | $P_\lambda(\beta) = \lambda\|\boldsymbol{\beta}\|_1$ | $\hat{\beta}_{lasso} = argmin\{\|\boldsymbol{y} - \boldsymbol{X\beta}\|_2^2 + \lambda\|\boldsymbol{\beta}\|_1\}$ |
| MCP [38] | $P_\lambda(\beta) = \begin{cases} \lambda\|\boldsymbol{\beta}\| - \dfrac{\|\boldsymbol{\beta}\|^2}{2\gamma}, & \|\boldsymbol{\beta}\| \le \gamma\lambda \\ \dfrac{1}{2}\gamma\lambda^2, & \|\boldsymbol{\beta}\| > \gamma\lambda \end{cases}$ | $\hat{\beta}_{MCP} = argmin\{\|\boldsymbol{y} - \boldsymbol{X\beta}\|_2^2 + P_\lambda(\beta)\}$ |
| SCAD [4] | $P_\lambda(\boldsymbol{\beta}) = \begin{cases} \lambda\|\boldsymbol{\beta}\|, & 0 \le \|\boldsymbol{\beta}\| \le \lambda \\ -\dfrac{(\|\boldsymbol{\beta}\|^2 - 2\gamma\lambda\|\boldsymbol{\beta}\| + \lambda^2)}{2(\gamma - 1)}, & \lambda < \|\boldsymbol{\beta}\| < \gamma\lambda \\ \dfrac{(\gamma + 1)\lambda^2}{2}, & \|\boldsymbol{\beta}\| \ge \gamma\lambda \end{cases}$ | $\hat{\beta}_{SCAD} = argmin\{\|\boldsymbol{y} - \boldsymbol{X\beta}\|_2^2 + P_\lambda(\beta)\}$ |
| enet [8] | $P_\lambda(\beta) = \lambda_1 \sum_{j=1}^{p} |\beta_j| + \lambda_2 \sum_{j=1}^{p} \beta_j^2$ | $\hat{\beta}_{enet} = argmin\{\|\boldsymbol{y} - \boldsymbol{X\beta}\|_2^2 + P_\lambda(\beta)\}$ |
| alasso [5] | $P_\lambda(\beta) = \lambda\|\boldsymbol{w\beta}\|_1$ | $\hat{\beta}_{alasso} = argmin\{\|\boldsymbol{y} - \boldsymbol{X\beta}\|_2^2 + \lambda\|\boldsymbol{w\beta}\|_1\}$ |

- Constructing a new feature matrix: The predictions of the base model are used as new features to construct a new feature matrix.

- Training meta-model: A meta-model is trained using the new feature matrix and the corresponding real labels.

- Making predictions: For test data, predictions are first generated using the base model, then these predictions are used to construct new feature matrices, and finally the meta-model is used to make the final predictions.

The benefit of stacking is its capacity to integrate the advantages of many models to enhance performance as a whole. It can take use of the variations between many models to provide predictions that are more reliable and accurate. However, there are certain limitations to stacking, such as selecting the appropriate base model, preventing overfitting, and handling input variations that may occur between various models.

In conclusion, stacking is an ensemble learning technique that combines the prediction outcomes of many fundamental models to get the final prediction result. It can enhance the model's performance and resilience and has applicability in many machine learning tasks, as shown in Fig 1.

It is important to keep in mind that the training set for the meta-learner is produced from the output of the base learner, and that utilizing the base learner's training set directly to produce the secondary training set might lead to severe overfitting. It is vital to logically partition the data utilization process in order to stop the two-layer learners from learning the same data again and to prevent the "overfitting" effect. The original training data set must be divided into three sub-datasets in accordance with the three base learners that were chosen, and it must be made sure that no two pieces of data ID overlap. For each base learner, 5-fold cross-validation is used, in which one block of data is used as the validation set and the remaining data as the training set. Fig 2 how each base learner may provide a prediction result, and how these three outcomes can ultimately be combined to create a new dataset that is the same size as the original. By realizing the feature transformation of all data from input features to output features

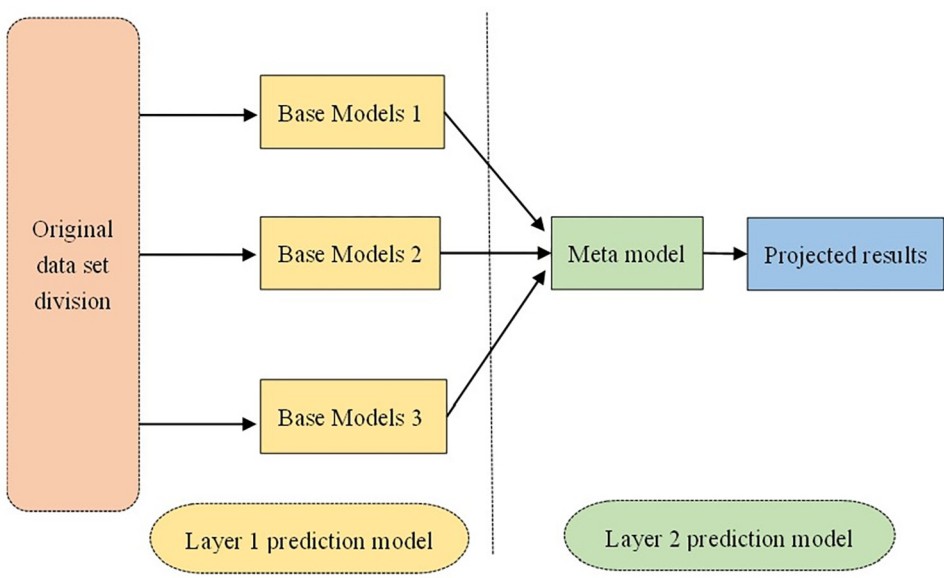

**Fig 1. Schematic diagram of Stacking-based integrated learning architecture.**

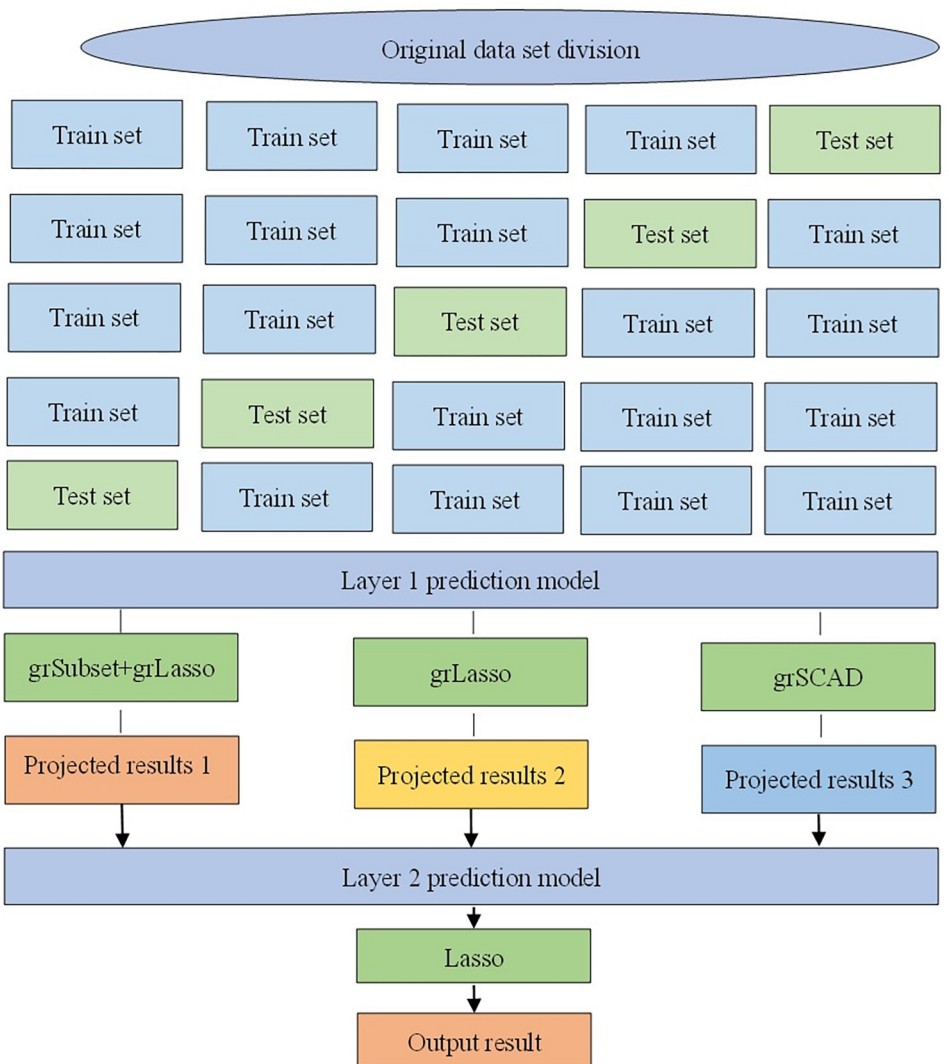

**Fig 2. Prediction method based on group structure penalty model under Stacking framework.**

and excluding the data blocks predicted by each base learner from their training, this configuration ensures that the data is only used once during the training of the model, effectively preventing overfitting.

The training process of the prediction method based on the cluster structure penalty model in the Stacking framework is as follows:

- The input features are trained for prediction using algorithms such as grSubset+grLasso, grLasso, and grSCAD to aid in feature selection;

- Analyze the error distribution of each algorithm and select the algorithm with larger differences as the layer 1 prediction model. Divide the original dataset and use cross-validation methods to optimize the optimal hyperparameters of each model;

- Produce a new dataset using the partitioned dataset to train the layer 1 prediction algorithm in Stacking separately and output the prediction results;

- Using the newly generated dataset, the layer 2 algorithm in Stacking is trained, based on which the Stacking integrated learning algorithm for multi-model fusion is trained.

## 2.7 Evaluation criteria

Regardless of whether a single Group Lasso method or an embedded integrated selection method is used, the model is selected and then the important variables are screened out. The variables selected by the model are then evaluated by using the model goodness-of-fit ($R^2$), the mean absolute error (*MAE*), and the root-mean-square error (*RMSE*) to ensure that they play an important role in the model. In regression problems, the three evaluation methods of $R^2$, *MAE* and *RMSE* are often used to evaluate the goodness of a regression model. Using these evaluation indexes can assess the performance of the model more comprehensively and objectively, and provide stronger support for the improvement and optimization of the model, and each of the three evaluation methods reflects the performance of the model from a different perspective, for example, $R^2$ measures the degree of fit of the model to the observed data, the *MAE* measures the mean absolute error between the predicted and observed values, and *RMSE* measures the root mean square error between the predicted and observed values, by using these three evaluation metrics can be corroborated with each other to mutually verify the performance of the model.

The goodness-of-fit index can be used to test the model's degree of fit to the data. The closer the value is to 1, the more stable the model is, and if the value is close to 0, then it means that the predicted value is not correlated with the observed value. $R^2$ (R-squared) is a measure of how well a model fits the observed data, and it can be used to assess the precision and reliability of the model. The calculation formula is as follows

$$R^2 = \frac{\sum_{i=1}^{n} (\hat{y}_i - \overline{y})^2}{\sum_{i=1}^{n} (y_i - \overline{y})^2} = 1 - \frac{\sum_{i=1}^{n} (y_i - \hat{y}_i)^2}{\sum_{i=1}^{n} (y_i - \overline{y})^2} \tag{5}$$

where $n$ denotes the number of samples, $y_i$ denotes the true value, $\hat{y}_i$ denotes the predicted value, and $\overline{y}$ denotes the mean of the true value.

The average absolute difference between the projected values and the real values is measured by the term "Mean Absolute Error" or MAE. The model matches the data better the smaller the value of MAE, the smaller the difference between the model's anticipated and real values. MAE does not take into account the relative orientations of the anticipated and real values; instead, it merely considers the distance between them. Since MAE directly takes the absolute values, it has the benefit of not being sensitive to outliers and not amplifying the impact of outliers. Although it is sensitive to outliers, it directly takes absolute values and does not exaggerate the impact of outliers. However, *MAE* also has some disadvantages, one of the main ones being that it does not take into account the square of the prediction error, which may lead to larger errors not being captured in some cases. The formula for calculating the *MAE* is as follows

$$MAE = \frac{1}{n} \sum_{i=1}^{n} |y_i - \hat{y}_i| \tag{6}$$

*RMSE* (Root Mean Square Error) is a measure of the root mean square error between the predicted value and the true value. The smaller the value of *RMSE* means the smaller the deviation between the predicted result and the true value of the model, i.e., the better the model is

fitted. Unlike the *MAE*, the *RMSE* is sensitive to large errors and it squares each error value before summing, so larger errors are magnified. The advantage of the *RMSE* is that it takes into account the squaring of the prediction error and is better able to capture larger errors. However, by taking the square, *RMSE* is more sensitive to outliers. If there are outliers in the data, it may result in a larger value of *RMSE*. The formula for calculating it is given below

$$RMSE = \sqrt{\frac{1}{n}\sum_{i=1}^{n}(y_i - \hat{y}_i)^2} \tag{7}$$

## 3 Simulation studies

In this section, we test the performance of the proposed algorithms mainly on simulated data. in order to compare the effects of different penalty functions on the variable selection effects discussed in this paper, simulation experiments are set up to compare different univariate selection and bivariate selection methods, including the grSubset+Ridge, grSubset+grLasso, grLasso, grMCP, grSCAD, lasso, MCP, SCAD, enet, alasso, and stacking+lasso algorithms. The model fit goodness of fit ($R^2$), mean absolute error (*MAE*), and root mean square error (*RMSE*) were used as indicators for evaluating the effectiveness of variable selection. Among them, the closer the model fit goodness of fit ($R^2$) is to 1, the more stable the model is, and the lower the other indicators are the better the model is. We designed five groups of simulation experiments for different cases of group variables, and the experiments included the number of non-empty groups, the number of significant variables within groups, the group size, and the distribution of significant variables between groups. Some of the model parameters used in this paper are set as follows:

**Group subset:** a grid of $\lambda_0$ chosen adaptively using the method of Proposition 1,where the first $\lambda_0$ sets all coefficients to zero;

**Group lasso:** a grid of $\lambda$ containing logarithmically spaced points between $\lambda^{max}$ and $\lambda^{min} = 10^{-4}\lambda_{max}$,where $\lambda^{max}$ is the smallest value that sets all coefficients to zero;;

**Group SCAD:** the same grid of $\lambda$ as above, and for each value of $\lambda$,a grid of the nonconvexity parameter $\gamma$ containing logarithmically spaced points between $\gamma^{max} = 100$ and $\gamma^{min} = 2+10^4$;

**Group MCP:** the same grid of $\lambda$ as above, and for each value of $\lambda$,a grid of the nonconvexity parameter $\gamma$ containing logarithmically spaced points between $\gamma^{max} = 100$ and $\gamma^{min} = 2+10^4$.

When using the grSubset+Ridge and grSubset+grLasso, refer to R package grpsel for specific parameter settings; Grids of 100 points are used for the primary tuning parameters ($\lambda_0, \lambda$) and grids of 30 points for the secondary tuning parameters ($\lambda_1,\gamma$).

The data for the simulation experiments in this paper were obtained from

$y_i = \sum_{g=1}^{G} X_g\beta_g + \varepsilon_i, i = 1,\cdots,n$, where $\varepsilon_i \sim^{i.i.d.} N(0,\sigma^2), \sigma^2 = 10$; The detailed setup of the simulation experiment is described below.

### Example1

Let $n = 200$ and $p = 500$, each group has 20 variables, for a total of 25 groups, $x_i \sim N(0,\Sigma)$, and the correlation coefficient of any two explanatory variables is $\rho = 0.5$. The covariance of the remaining variables is 0. The true regression coefficient $\boldsymbol{\beta}$ is specified as:

$\boldsymbol{\beta} = \left[\underbrace{1,1,\cdots,1}_{group1}, \underbrace{0,0,\cdots,0}_{group2}, \underbrace{1,1,\cdots1}_{group3}, \underbrace{0,0,\cdots,0}_{group4}, \cdots, \underbrace{1,1,\cdots,1}_{group25}\right]^T$, the simulation results are shown in the table below.

In Table 2, we compare the method proposed in this paper with the rest of the group structure penalty function models. we can conclude that for the case of $n < p$, when the sample size

**Table 2. Comparison of the results of simulation experiment 1 for several group structure penalty models.**

| Model | $R^2$ | RMSE | MAE |
|---|---|---|---|
| grSubset+Ridge | 0.936 | 40.040 | 32.579 |
| grSubset+grLasso | 0.939 | 39.167 | 32.194 |
| grLasso | 0.987 | 75.886 | 59.828 |
| grMCP | 0.939 | 39.193 | 30.546 |
| grSCAD | 0.986 | 78.126 | 61.636 |
| lasso | 0.962 | 37.042 | 28.730 |
| MCP | 0.886 | 55.271 | 44.360 |
| SCAD | 0.936 | 52.737 | 41.970 |
| enet | 0.990 | 141.038 | 111.869 |
| alasso | 0.921 | 45.110 | 36.406 |
| stacking+lasso | **0.973** | **27.127** | **21.193** |

n = 200 and the number of variables p = 500, the model fit of the enet model is better, but the RMSE and MAE errors are larger; whereas, the stacking+lasso method is in the middle-upper level of the model fit compared to the other several models, where the value of the index reaches the minimum for the RMSE and the MAE compared to the other models, which indicates that the model's true and predicted values have less errors.

**Example2.** Let $n = 200$ and $p = 1000$, each group has 20 variables and there are 50 groups, $x_i \sim N(0,\Sigma)$, and the correlation coefficient of any two explanatory variables is $\rho = 0.5$. The covariance of the remaining variables is 0. The true regression coefficient $\boldsymbol{\beta}$ is specified as:

$$\boldsymbol{\beta} = \left[\underbrace{1,1,\cdots,1}_{group1}, \underbrace{0,0,\cdots,0}_{group2}, \underbrace{1,1,\cdots 1}_{group3}, \underbrace{0,0,\cdots,0}_{group4}, \cdots, \underbrace{0,\cdots,0}_{group50}\right]^T,$$ the simulation results are

shown in the table below.

In Table 3, for the case of $n < p$, when the sample size $n = 200$ and the number of variables $p = 1000$, the model fit of the enet model is better, but the *RMSE* and *MAE* errors are larger; while the stacking+lasso method is in the middle-upper level of the model fit compared to the other several models, in which the value of the *RMSE* and the *MAE* is the smallest compared to the other several models, which indicates that the model's real and predicted values have small errors, and the model fit of this method is gradually improving with the increase in the number of variables, meanwhile the values of the *RMSE* and the *MAE* are also rising.

**Table 3. Comparison of the results of simulation experiment 2 for several group structure penalty models.**

| Model | $R^2$ | RMSE | MAE |
|---|---|---|---|
| grSubset+Ridge | 0.939 | 73.305 | 59.054 |
| grSubset+grLasso | 0.942 | 71.331 | 57.747 |
| grLasso | 0.986 | 159.936 | 131.542 |
| grMCP | 0.937 | 76.423 | 62.216 |
| grSCAD | 0.987 | 159.983 | 131.559 |
| lasso | 0.975 | 56.084 | 43.897 |
| MCP | 0.907 | 87.124 | 69.862 |
| SCAD | 0.976 | 65.467 | 49.895 |
| enet | 0.992 | 276.974 | 224.425 |
| alasso | 0.943 | 76.180 | 61.773 |
| stacking+lasso | **0.975** | **48.622** | **39.388** |

**Table 4. Comparison of the results of simulation experiment 3 for several group structure penalty models.**

| Model | $R^2$ | RMSE | MAE |
|---|---|---|---|
| grSubset+Ridge | 0.934 | 109.583 | 86.280 |
| grSubset+grLasso | 0.939 | 104.601 | 80.775 |
| grLasso | 0.985 | 235.129 | 189.434 |
| grMCP | 0.955 | 91.063 | 72.428 |
| grSCAD | 0.984 | 241.740 | 194.482 |
| lasso | 0.965 | 118.773 | 92.327 |
| MCP | 0.869 | 154.296 | 126.395 |
| SCAD | 0.961 | 131.099 | 103.294 |
| enet | 0.990 | 415.292 | 329.605 |
| alasso | 0.915 | 133.707 | 106.672 |
| stacking+lasso | **0.973** | **96.495** | **74.367** |

**Example3.** Let $n$ = 200 and $p$ = 1500, each group has 20 variables and there are 75 groups, $x_i \sim N(0, \Sigma)$, and the correlation coefficient of any two explanatory variables is $\rho$ = 0.5. The covariance of the remaining variables is 0. The true regression coefficient $\boldsymbol{\beta}$ is specified as:

$$\boldsymbol{\beta} = \left[\underbrace{1, 1, \cdots, 1}_{group1}, \underbrace{0, 0, \cdots, 0}_{group2}, \underbrace{1, 1, \cdots 1}_{group3}, \underbrace{0, 0, \cdots, 0}_{group4}, \cdots, \underbrace{1, 1, \cdots, 1}_{group75}\right]^T,$$ the simulation results

are shown in the table below.

In Table 4, for the case of $n < p$, when the sample size $n$ = 200 and the number of variables $p$ = 1500, the model fit of the enet model is better, and the model fit of the model decreases with the increase of the number of variables, but the error values of the *RMSE* and the *MAE* are still larger; while the stacking+lasso method is in the middle to the upper level, where the *RMSE* and the *MAE* are the smallest values compared to other models, indicating that the model has less error in the true value and the predicted value, and that the model has less error in the real value and the predicted value. upper level, where the value of *RMSE* and *MAE* is the smallest compared to several other models, indicating that the model's true and predicted values have smaller errors, with the increase of the number of variables, the method's model fit also decreases, and at the same time, the values of *RMSE* and *MAE* are also increasing.

**Example4.** Let $n$ = 200 and $p$ = 2000, each group has 20 variables and there are 100 groups, $x_i \sim N(0, \Sigma)$, and the correlation coefficient of any two explanatory variables is $\rho$ = 0.5. The covariance of the remaining variables is 0. The true regression coefficient $\beta$ is specified as:

$$\boldsymbol{\beta} = \left[\underbrace{1, 1, \cdots, 1}_{group1}, \underbrace{0, 0, \cdots, 0}_{group2}, \underbrace{1, 1, \cdots 1}_{group3}, \underbrace{0, 0, \cdots, 0}_{group4}, \cdots, \underbrace{0, 0, \cdots, 0}_{group100}\right]^T,$$ the simulation results

are shown in the table below.

In Table 5, for the case of $n < p$, when the sample size $n$ = 200 and the number of variables $p$ = 2000, the model fit of the enet model is better, but the error values of the *RMSE* and *MAE* are still larger; while the stacking+lasso method has an intermediate to upper level of model fit compared to several other models, where the value of the *RMSE* and *MAE* is the smallest compared to several other models, which indicates that the model's true and predicted values have a smaller error, and the model fit of this method decreases with the increase in the number of variables, and at the same time the values of the *RMSE* and the *MAE* are also rising.

**Example5.** Let $x_i \sim N(0, \Sigma)$, and the correlation coefficient of any two explanatory variables is $\rho$ = 0.5, and the covariance of the rest of the variables is 0. The true regression coefficients $\boldsymbol{\beta}$ repeat the setup of the simulation experiment above, and the simulation results are shown in the following Table 6.

**Table 5. Comparison of the results of simulation experiment 4 for several group structure penalty models.**

| Model | $R^2$ | RMSE | MAE |
|---|---|---|---|
| grSubset+Ridge | 0.929 | 145.855 | 112.437 |
| grSubset+grLasso | 0.930 | 145.256 | 111.937 |
| grLasso | 0.985 | 303.653 | 248.411 |
| grMCP | 0.921 | 154.063 | 128.208 |
| grSCAD | 0.985 | 295.929 | 241.964 |
| lasso | 0.969 | 131.241 | 104.894 |
| MCP | 0.831 | 224.844 | 183.346 |
| SCAD | 0.966 | 164.393 | 130.490 |
| enet | 0.991 | 526.947 | 431.933 |
| alasso | 0.888 | 208.119 | 163.305 |
| stacking+lasso | **0.970** | **110.398** | **90.160** |

In order to find the best combination of base learners in the Stacking model, the model with better model fit goodness-of-fit ($R^2$), mean absolute error (MAE) and root mean square error (RMSE) in the simulation experiments 1–4 is selected as the base learner for the Stacking model; as can be seen from the Table 6, the algorithmic model combinations such as grSubset +grLasso, grLasso, grSCAD and lasso and the algorithmic model combinations of grSubset+-grLasso, grLasso, and grSCAD as the base learner, the former has a larger model goodness-of-fit ($R^2$), mean absolute error (MAE), and root-mean-square error (RMSE) relative to the latter with the increase in the number of variables, which indicates that the latter model combination leads to a better model; and in the same dimension, the model fit goodness of fit ($R^2$), mean absolute error (MAE), and root mean square error (RMSE) are also better for the latter compared to the former.

To further explore the performance of the method proposed in this paper in the case of large samples in high dimensions, the experimental setup is designed to be similar to that of the previous simulation experiments, except that the sample size $n$ and dimension $p$ are different, and the experimental results are shown in Table 7.

With the increase of the sample size, we can observe from the table that the StackingGroup algorithm is optimal in all aspects of the indicators in the case of sample size $n = 1000$ and dimension $p = 100$; in the same dimension, the algorithm proposed in this paper, compared to other algorithms, the three evaluation indexes give better results, which is enough to show that the algorithm in the selection of variables and the model prediction, the algorithm's prediction accuracy and modeling error will not be inferior to other algorithms.

**Table 6. Comparison of the results of simulation experiment 5 of the Stacking algorithm with different base model combination approaches.**

| Model | Model Combination | Sample Size | The Evaluation Index | | |
|---|---|---|---|---|---|
| | | | $R^2$ | RMSE | MAE |
| **Stacking 1** | **Base Models:**grSubset+grLasso, grLasso, grSCAD, lasso | $p = 500, group = 25$ | 0.977 | 26.275 | 20.485 |
| | | $p = 1000, group = 50$ | 0.979 | 46.281 | 36.863 |
| | | $p = 1500, group = 75$ | 0.974 | 93.837 | 72.744 |
| | **Meta Model:**lasso | $p = 2000, group = 100$ | 0.979 | 100.007 | 81.349 |
| **Stacking 2** (Stacking-Group) | **Base Models:**grSubset+grLasso, grLasso, grSCAD | $p = 500, group = 25$ | 0.990 | 15.371 | 12.251 |
| | | $p = 1000, group = 50$ | 0.987 | 34.740 | 26.214 |
| | **Meta Model:**lasso | $p = 1500, group = 75$ | 0.990 | 46.411 | 38.874 |
| | | $p = 2000, group = 100$ | 0.988 | 67.448 | 51.167 |

**Table 7. Comparison of simulation results of different group structure penalty models in the case of high dimensional large samples.**

| Sample Size | Model | $R^2$ | RMSE | MAE |
|---|---|---|---|---|
| n = 500 | grSubset+grLasso | 0.905 | 12.830 | 10.406 |
|  | grLasso | 0.925 | 11.702 | 9.690 |
|  | grSCAD | 0.904 | 11.972 | 9.734 |
|  | grSubset+Ridge | 0.901 | 12.148 | 9.884 |
|  | grMCP | 0.918 | 11.151 | 8.944 |
| p = 100 | lasso | 0.921 | 10.947 | 8.870 |
|  | MCP | 0.913 | 11.393 | 9.339 |
|  | SCAD | 0.915 | 11.287 | 9.426 |
|  | alasso | 0.912 | 11.516 | 9.455 |
|  | StackingGroup | **0.919** | **11.020** | **8.947** |
| n = 1000 | grSubset+grLasso | 0.907 | 12.048 | 9.896 |
|  | grLasso | 0.932 | 9.614 | 7.555 |
|  | grSCAD | 0.908 | 10.977 | 8.686 |
|  | grSubset+Ridge | 0.906 | 11.069 | 8.988 |
|  | grMCP | 0.930 | 9.535 | 7.434 |
| p = 100 | lasso | 0.929 | 9.558 | 7.490 |
|  | MCP | 0.922 | 10.130 | 8.015 |
|  | SCAD | 0.925 | 9.904 | 7.813 |
|  | alasso | 0.925 | 9.853 | 7.677 |
|  | StackingGroup | **0.932** | **9.399** | **7.318** |
| n = 1500 | grSubset+grLasso | 0.896 | 12.985 | 10.350 |
|  | grLasso | 0.918 | 10.866 | 8.579 |
|  | grSCAD | 0.912 | 10.972 | 8.643 |
|  | grSubset+Ridge | 0.894 | 12.083 | 9.330 |
|  | grMCP | 0.917 | 10.649 | 8.435 |
| p = 100 | lasso | 0.915 | 10.800 | 8.587 |
|  | MCP | 0.914 | 10.872 | 8.616 |
|  | SCAD | 0.914 | 10.872 | 8.612 |
|  | alasso | 0.913 | 10.934 | 8.707 |
|  | StackingGroup | **0.918** | **10.642** | **8.437** |
| n = 2000 | grSubset+grLasso | 0.899 | 13.133 | 10.350 |
|  | grLasso | 0.925 | 10.829 | 8.664 |
|  | grSCAD | 0.922 | 10.934 | 8.695 |
|  | grSubset+Ridge | 0.899 | 12.437 | 9.815 |
|  | grMCP | 0.925 | 10.677 | 8.512 |
| p = 100 | lasso | 0.924 | 10.737 | 8.516 |
|  | MCP | 0.924 | 10.801 | 8.584 |
|  | SCAD | 0.925 | 10.703 | 8.509 |
|  | alasso | 0.923 | 10.817 | 8.612 |
|  | StackingGroup | **0.925** | **10.667** | **8.478** |

## 4 Real data

### 4.1 Data set: Brithwt

The Brithwt dataset, which collects data on the risk factors for low birth weight in infants and young children, is an extension of the Brithwt dataset collected at the Springfield Medical Center in Springfield, Massachusetts, in 1986, and can be loaded using the R package grpreg,

"Birthwt". The data set has a total of 189 infant weights and 8 predictors related to pregnant mothers, including the mother's age (age), mother's weight during the last menstrual period (lwt), mother's race (race), previous preterm births (ptl), number of physician visits within 3 months (ftv), smoking during pregnancy (smoke), history of hypertension (ht), and uterine stimulation (ui). Among them, age and weight variables are continuous variables, and the remaining explanatory variables are categorical variables. Because the mother's age and weight may have a non-linear relationship, the first-order, second-order, and third-order polynomials of the age variable and the weight variable are used to replace the original variables, that is, the age variable is divided into age1, age2, and age3 according to different age groups. Three indicator variables, lwt variable are divided into lwt1, lwt2, and lwt3 according to weight grade. The ftv variable is divided into three indicator variables ft1, ft2, and ftm according to the number of visits. The data set has a total of 8 groups. For this data set, we hope to find out the important factors related to the weight of the newborn through the group variable selection integration algorithm.

## 4.2 Experiment environment

The experimental environment includes both software and hardware. On the software side, the methods are implemented using the R language, the version of which is R.4.2.2. On the hardware side, the system runs on a 12th Gen Intel(R) Core(TM) i7-12700H 2.30 GHz. The datasets generated and/or analyzed during the current study are not publicly available due all experimental and empirical data in this paper are generated and pulled from R.4.2.2 simulations but are available from the corresponding author on reasonable request.

## 4.3 Results

In order to verify the effectiveness of the proposed method in this paper, grSubset+grLasso, grLasso, grSCAD, and other models in the simulation experiments are introduced to compare with the proposed method, and we randomly divide the dataset into a training set (70% of the observations) and a test set (30% of the observations), in which the proposed method adopts a 5-fold cross-validation and obtains the results of the comparison experiments in Table 8, in which the optimal performance indexes are bold; the important variables screened out by each algorithm are given in Table 9, and in order to have a more intuitive understanding of each algorithm's performance in the dataset, the results of the plotted images are shown in Fig 3.

As we can see, combining the three evaluation metrics in Table 8 and Fig 3, the method proposed in this paper has a higher model fit $R^2$, and both the root mean square error ($RMSE$) and

**Table 8. Comparison of modeling results of different models for Brithwt dataset.**

| Model | $R^2$ | $RMSE$ | $MAE$ |
|---|---|---|---|
| grSubset+grLasso | 0.075 | 0.751 | 0.579 |
| grLasso | 0.266 | 0.704 | 0.544 |
| grSCAD | 0.177 | 0.719 | 0.562 |
| grSubset+Ridge | 0.204 | 0.705 | 0.551 |
| grMCP | 0.193 | 0.706 | 0.553 |
| lasso | 0.255 | 0.709 | 0.549 |
| MCP | 0.194 | 0.715 | 0.557 |
| SCAD | 0.235 | 0.692 | 0.539 |
| StackingGroup | **0.301** | **0.668** | **0.508** |

**Note:** StackingGroup (**Base Models:**grSubset+grLasso, grLasso, grSCAD; **Meta Model:**lasso)

**Table 9. Comparison of variable selection results of different models for Brithwt dataset.**

| Variable | grSubset+grLasso | grLasso | grSCAD | grSubset+Ridge | grMCP | lasso | MCP | SCAD | StackingGroup |
|---|---|---|---|---|---|---|---|---|---|
| age1 | | | | | | | | | |
| age2 | | | | | | | | | |
| age3 | | | | | | | | | |
| lwt1 | | ✓ | ✓ | | | ✓ | ✓ | ✓ | ✓ |
| lwt2 | | ✓ | ✓ | | | ✓ | | | ✓ |
| lwt3 | | ✓ | ✓ | | | | | | ✓ |
| white | | ✓ | ✓ | ✓ | ✓ | ✓ | ✓ | ✓ | ✓ |
| black | | ✓ | ✓ | ✓ | ✓ | | | | ✓ |
| smoke | ✓ | ✓ | ✓ | ✓ | ✓ | ✓ | ✓ | ✓ | ✓ |
| ptl1 | | ✓ | ✓ | | ✓ | ✓ | | ✓ | ✓ |
| ptl2m | | ✓ | ✓ | | ✓ | | | | ✓ |
| ht | | ✓ | | | | | | | ✓ |
| ui | ✓ | ✓ | ✓ | ✓ | ✓ | ✓ | ✓ | ✓ | ✓ |
| ftv1 | | | | | | | | | |
| ftv2 | | | | | | | | | |
| ftv3m | | | | | | | | | |
| ZJ | 2 | 9 | 9 | 4 | 6 | 6 | 4 | 5 | 9 |

Note: ZJ is a count of the number of variables selected by a particular model, and "✓" indicates that the indicator was selected by that model.

the mean absolute error ($MAE$) are smaller than those of several other models, thus it can be shown that the integrated algorithm has a better model fit and model error of the integrated model compared to a single model. will be better compared to a single model. In conclusion,

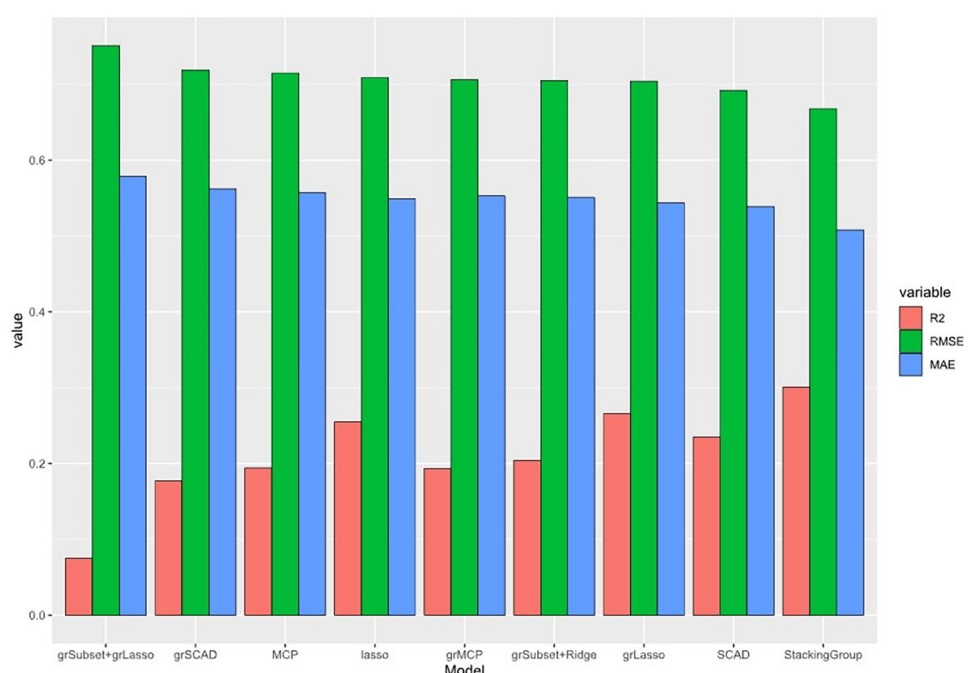

**Fig 3. Evaluation results of all methods for modeling the Brithwt dataset: Model fit $R^2$, root mean square error ($RMSE$) and mean absolute error ($MAE$).**

StackingGroup produces excellent performance on this dataset, and in real-world applications, we look forward to using the integrated model to solve problems in practice.

In Table 9, the grLasso, grSCAD, and StackingGroup algorithms select 9 explanatory variables from 16 explanatory variables and 8 groups. The grMCP and lasso algorithms select six explanatory variables; the SCAD algorithm selects five explanatory variables; grSubset + Ridge and MCP selected 4 explanatory variables; the grSubset + grLasso algorithm selects two explanatory variables. Therefore, the method proposed in this paper is effective and suitable for real data with group structure.

## 5 Conclusion

Variable selection is a hot research issue in the field of statistics, in this paper, we propose a multi-algorithm fusion model (StackingGroup) based on the Stacking integrated learning framework, which is constructed by using grSubset+grLasso, grLasso, and grSCAD as the base learners and lasso as the meta-learners, the algorithm proposed can solve the variable selection problem of high-dimensional group structure data, and it features high recognition of important inter-group variables and important intra-group variables; the effectiveness and stability of the method is verified through numerical simulation and empirical analysis, thus indicating that the algorithm has certain practical application value. However, it still has some limitations, for example, due to the multiple algorithm fusion strategy adopted by the algorithm, the computation of the algorithm changes with the model complexity of the base learner and the meta-learners; algorithm optimization and parallel computation may be required when facing the computation of large-scale data. In our next work, we will continue to investigate how to perform bilayer variable selection and its application to disease prediagnosis and economic forecasting under the Stacking integrated learning framework when the response variable is discrete. In addition, optimization of algorithms, computation, and processing of large-scale datasets are also future research directions in this area.

## Supporting information

**S1 Data.**
(ZIP)

## Author Contributions

**Conceptualization:** Dongsheng Li, Anfei Luo.

**Formal analysis:** Dongsheng Li, Anfei Luo.

**Methodology:** Chunyan Pan, Jing Zhao, Anfei Luo.

**Validation:** Chunyan Pan.

**Writing – original draft:** Dongsheng Li.

**Writing – review & editing:** Dongsheng Li, Chunyan Pan, Jing Zhao, Anfei Luo.

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
