## [Decision Letter · Decision Letter 0]

7 Nov 2023

PONE-D-23-35145A Penalized Variable Selection Ensemble Algorithm for High-Dimensional Group-Structured DataPLOS ONE

Dear Dr. Li,

Thank you for submitting your manuscript to PLOS ONE. After careful consideration, we feel that it has merit but does not fully meet PLOS ONE’s publication criteria as it currently stands. Therefore, we invite you to submit a revised version of the manuscript that addresses the points raised during the review process.

We look forward to receiving your revised manuscript.

Kind regards,

Nasir Ayub, Ph.D.

Academic Editor

PLOS ONE

Journal Requirements:

“This work was supported by The Guizhou Provincial Department of Education's Youth Growth Project Fund [No. Qian Jiaoji [2022] 380, No. Qian Jiaoji [2022] 377, No. Qian Jiaoji [2022] 378, and No. Qian Jiaohe KY [2021] 282]; and in part by the Educational Department of Guizhou under Grant [No.KY[2019]067].”

“This work was supported by The Guizhou Provincial Department of Education's Youth Growth Project Fund [No. Qian Jiaoji [2022] 380, No. Qian Jiaoji [2022] 377, No. Qian Jiaoji [2022] 378, and No. Qian Jiaohe KY [2021] 282]; and in part by the Educational Department of Guizhou under Grant [No.KY[2019]067].”

“This work was supported by The Guizhou Provincial Department of Education's Youth Growth Project Fund [No. Qian Jiaoji [2022] 380, No. Qian Jiaoji [2022] 377, No. Qian Jiaoji [2022] 378, and No. Qian Jiaohe KY [2021] 282]; and in part by the Educational Department of Guizhou under Grant [No.KY[2019]067].”

Additional Editor Comments (if provided):

Dear Dr. Dongsheng Li,

Thank you for your submission to PLOS One. We appreciate your contribution to our journal. However, before resubmission, we kindly request that you carefully address the reviewers' comments and suggestions.

Regards,

Dr. Nasir Ayub

Academic Editor

Reviewers' comments:

Reviewer's Responses to Questions

**Comments to the Author**

1. Is the manuscript technically sound, and do the data support the conclusions?

Reviewer #1: No

Reviewer #2: Yes

2. Has the statistical analysis been performed appropriately and rigorously? 

Reviewer #1: No

Reviewer #2: Yes

3. Have the authors made all data underlying the findings in their manuscript fully available?

Reviewer #1: No

Reviewer #2: Yes

4. Is the manuscript presented in an intelligible fashion and written in standard English?

Reviewer #1: Yes

Reviewer #2: Yes

5. Review Comments to the Author

Reviewer #1: Your methodology, involving comprehensive simulation experiments comparing the StackingGroup model with various algorithms under different group variable scenarios, is commendable. However, for improved reader comprehension, providing context for the evaluation metrics (R², MAE, RMSE) is advised. Additionally, a deeper explanation of data generation and the unique contributions of StackingGroup in various scenarios, along with the incorporation of visual aids, will enhance the presentation's clarity and value. Furthermore, linking the results to practical implications for researchers and practitioners would enrich the paper's real-world relevance.

Reviewer #2: The article is quite good, however, some concerns needs to be addressed:

While the paper presents a novel algorithm, it would be beneficial to discuss the limitations and potential challenges associated with implementing the StackingGroup algorithm. Addressing practical issues or constraints in real-world applications would provide a more comprehensive perspective.

The paper mentions that the algorithm "outperformed other prediction methods," but it would be more informative if the authors could provide a direct comparison with specific existing methods, including their strengths and weaknesses in different scenarios. This would help the readers understand where StackingGroup excels and where it might fall short.

In terms of experimental evaluation, the paper mainly focuses on statistical metrics (R2, RMSE, MAE). It would be valuable to provide additional context or case studies to illustrate how the algorithm's enhanced predictive accuracy can lead to meaningful insights or applications in various domains. Real-life use cases could strengthen the practical relevance of the algorithm.

The paper's clarity is commendable, but a more detailed explanation of the algorithm's inner workings, such as its convergence properties, optimization strategies, and computational complexity, would be beneficial. Understanding these aspects can help researchers and practitioners decide when and how to apply the StackingGroup algorithm effectively.

It's essential to consider the algorithm's generalizability. Are there specific types of data or situations where it may not perform as well? Discussing the algorithm's limitations and potential failure modes would provide a more balanced view.

The paper could be improved by including visual aids, such as diagrams or charts, to illustrate the StackingGroup algorithm's workflow or the outcomes of the experiments. Visual representations can make complex concepts more accessible.

While the paper provides a strong argument for the StackingGroup algorithm's superior performance, it should also address any overfitting concerns. A discussion of how the algorithm handles overfitting or whether there are cases where it might struggle with noisy data would enhance its credibility.

Consider discussing the scalability of the algorithm, especially when applied to large datasets. If there are computational limitations or challenges related to handling big data, it's important to acknowledge them.

The paper's conclusion could be strengthened by summarizing the key contributions, highlighting the practical implications of the StackingGroup algorithm, and suggesting potential directions for future research in this area.

Lastly, the author ensure that all mathematical notations and equations are clear and consistently formatted throughout the paper. Clarity and readability are essential for the paper's success.

6. PLOS authors have the option to publish the peer review history of their article (what does this mean?). If published, this will include your full peer review and any attached files.

Reviewer #1: No

Reviewer #2: No

---

## [Author Response · Author response to Decision Letter 0]

17 Nov 2023

Funding

This work was supported by The Guizhou Provincial Department of Education's Youth Growth Project Fund [No. Qian Jiaoji [2022] 380, No. Qian Jiaoji [2022] 377, No. Qian Jiaoji [2022] 378, and No. Qian Jiaohe KY [2021] 282]; and in part by the Educational Department of Guizhou under Grant [No.KY[2019]067].

Competing interests

The authors have declared that no competing interests exist.

---

## [Decision Letter · Decision Letter 1]

19 Dec 2023

A Penalized Variable Selection Ensemble Algorithm for High-Dimensional Group-Structured Data

PONE-D-23-35145R1

Dear Dr. Luo,

We’re pleased to inform you that your manuscript has been judged scientifically suitable for publication and will be formally accepted for publication once it meets all outstanding technical requirements.

Kind regards,

Nasir Ayub, Ph.D.

Academic Editor

PLOS ONE

Additional Editor Comments (optional):

Reviewers' comments:

Reviewer's Responses to Questions

**Comments to the Author**

1. If the authors have adequately addressed your comments raised in a previous round of review and you feel that this manuscript is now acceptable for publication, you may indicate that here to bypass the “Comments to the Author” section, enter your conflict of interest statement in the “Confidential to Editor” section, and submit your "Accept" recommendation.

Reviewer #1: All comments have been addressed

Reviewer #2: All comments have been addressed

2. Is the manuscript technically sound, and do the data support the conclusions?

Reviewer #1: Yes

Reviewer #2: Yes

3. Has the statistical analysis been performed appropriately and rigorously? 

Reviewer #1: Yes

Reviewer #2: Yes

4. Have the authors made all data underlying the findings in their manuscript fully available?

Reviewer #1: Yes

Reviewer #2: (No Response)

5. Is the manuscript presented in an intelligible fashion and written in standard English?

Reviewer #1: Yes

Reviewer #2: Yes

6. Review Comments to the Author

Reviewer #1: (No Response)

Reviewer #2: The authors have addressed all the reviews and incorporated the changes. Now, the article is fit for acceptance.

7. PLOS authors have the option to publish the peer review history of their article (what does this mean?). If published, this will include your full peer review and any attached files.

Reviewer #1: No

Reviewer #2: No

---

## [Editor Report · Acceptance letter]

23 Jan 2024

PONE-D-23-35145R1 

PLOS ONE

Dear Dr. Luo, 

I'm pleased to inform you that your manuscript has been deemed suitable for publication in PLOS ONE. Congratulations! Your manuscript is now being handed over to our production team.

Kind regards, 

on behalf of

Dr. Nasir Ayub 

Academic Editor

PLOS ONE